# Altered microRNA Transcriptome in Cultured Human Airway Cells upon Infection with SARS-CoV-2

**DOI:** 10.3390/v15020496

**Published:** 2023-02-10

**Authors:** Idrissa Diallo, Rajesh Abraham Jacob, Elodie Vion, Robert A. Kozak, Karen Mossman, Patrick Provost

**Affiliations:** 1CHU de Québec Research Center/CHUL Pavilion, Department of Microbiology, Infectious Diseases and Immunology, Faculty of Medicine, Université Laval, Quebec City, QC G1V 0A6, Canada; 2McMaster Immunology Research Centre, McMaster University, Hamilton, ON L8S 4K1, Canada; 3Department of Medicine, McMaster University, Hamilton, ON L8S 4K1, Canada; 4Division of Microbiology, Department of Laboratory Medicine & Molecular Diagnostics, Sunnybrook Health Sciences Centre, Toronto, ON M4N 3M5, Canada; 5M.G. DeGroote Institute for Infectious Disease Research, McMaster University, Hamilton, ON L8S 4K1, Canada

**Keywords:** SARS-CoV-2, RNA-Seq, miRNA, miR-1246, ACE2, Calu-3

## Abstract

Numerous proteomic and transcriptomic studies have been carried out to better understand the current multi-variant SARS-CoV-2 virus mechanisms of action and effects. However, they are mostly centered on mRNAs and proteins. The effect of the virus on human post-transcriptional regulatory agents such as microRNAs (miRNAs), which are involved in the regulation of 60% of human gene activity, remains poorly explored. Similar to research we have previously undertaken with other viruses such as Ebola and HIV, in this study we investigated the miRNA profile of lung epithelial cells following infection with SARS-CoV-2. At the 24 and 72 h post-infection time points, SARS-CoV-2 did not drastically alter the miRNome. About 90% of the miRNAs remained non-differentially expressed. The results revealed that miR-1246, miR-1290 and miR-4728-5p were the most upregulated over time. miR-196b-5p and miR-196a-5p were the most downregulated at 24 h, whereas at 72 h, miR-3924, miR-30e-5p and miR-145-3p showed the highest level of downregulation. In the top significantly enriched KEGG pathways of genes targeted by differentially expressed miRNAs we found, among others, MAPK, RAS, P13K-Akt and renin secretion signaling pathways. Using RT-qPCR, we also showed that SARS-CoV-2 may regulate several predicted host mRNA targets involved in the entry of the virus into host cells (ACE2, TMPRSS2, ADAM17, FURIN), renin–angiotensin system (RAS) (Renin, Angiotensinogen, ACE), innate immune response (IL-6, IFN1β, CXCL10, SOCS4) and fundamental cellular processes (AKT, NOTCH, WNT). Finally, we demonstrated by dual-luciferase assay a direct interaction between miR-1246 and ACE-2 mRNA. This study highlights the modulatory role of miRNAs in the pathogenesis of SARS-CoV-2.

## 1. Introduction

Two years after the outbreak of the coronavirus disease 19 (COVID-19) pandemic, the disease remains a major public health concern with millions of deaths and hospitalizations reported [1]. Its infectious agent, severe acute respiratory syndrome coronavirus 2 (SARS-CoV-2), a new betacoronavirus of the coronaviridae family [2], has not revealed all its secrets despite the global efforts of the scientific community and political leaders.

Although the availability of vaccines and drugs which help to reduce deaths and hospitalizations [3] have given new hope in the fight against the pandemic, we are still threatened by the emergence of concerning variants. It is therefore important to deepen our understanding of SARS-CoV-2, to better characterize the molecular mechanisms that underlie its zoonotic transfer, replication, persistence, transmission and lethality [4].

SARS-CoV-2 is a positive-sense single-stranded RNA virus and is composed of several accessory proteins and four structural proteins including envelope (E), membrane (M), nucleocapsid (N) and spike (S) [5]. The latter, spike (S), mediates the entry of the virus into host cells by interacting with angiotensin-converting enzyme 2 (ACE2) and other proteases such as transmembrane serine protease 2 (TMPRSS2), ADAM metallopeptidase domain 17 (ADAM17) and Furin [6,7,8]. The virus leads to an acute respiratory distress syndrome, systemic hyperinflammation (leading to a “cytokine storm”) and multiple organ failure [9].

In addition to being poorly understood, the molecular mechanisms behind the immunopathogenesis of COVID19 is often investigated only from the perspective of protein macromolecules [10]. The role of non-coding RNAs in COVID-19, in particular microRNAs (miRNAs), is not fully comprehended. However, it is now acknowledged that miRNAs influence more than 60% of a given host transcriptome [11].

miRNAs are 19 to 24 nucleotide (nt) RNAs that induce posttranscriptional repression of their mRNA targets (for review, see [12]) and have been viewed as a possible approach to prevent SARS-CoV-2 replication [13]. Nevertheless, miRNAs may, under certain conditions, be the Achilles heel of the immune defense that SARS-CoV-2 would exploit [14]. This functional duality illustrates the critical role of miRNAs in viral infections [15] and their potential utility as informative biomarkers for disease diagnosis and prognosis [16,17].

In the ongoing pandemic, timely reports (about thirty quality publications released before the end of 2020) provided evidence on the impact of miRNAs on SARS-CoV-2 infection and COVID-19 [18]. Subsequently, further studies performed comprehensive miRNA profiling in various tissues [19,20,21]. However, in the context of COVID-19, few studies have been conducted to explore the global profile of miRNAs (miRNome) by deep sequencing technologies, especially in the lung, the primary organ targeted by the virus [22,23,24].

In the current study, we profiled the miRNome at three time points (24 h, 48 h and 72 h) following SARS-CoV-2 infection of lung epithelial cells by RNA sequencing (RNA-Seq). Our results reveal that 90% of the miRNAs were non-differentially expressed and that only a limited pool of miRNAs (69 miRNAs at 24 h and 25 miRNAs at 72 h) was specifically modulated by the virus. In the top significantly enriched KEGG pathways of genes targeted by differentially expressed miRNAs, we found, among others, MAPK, RAS, P13K-Akt and renin secretion signaling pathways. Using RT-qPCR, we also showed that SARS-CoV-2 may regulate several predicted host mRNA targets involved in the entry of the virus into the host cells (ACE2, TMPRSS2, ADAM17, Furin), renin–angiotensin–aldosterone system (RAAS) (Renin, Angiotensinogen, ACE), innate immune response (IL-6, IFN1β, CXCL10, SOCS4) and fundamental cellular processes (AKT, NOTCH, WNT). RNA-Seq data highlighted a significant upregulation of miR-1246 (confirmed by RT-qPCR), which was competent in a dual-luciferase assay to directly modulate ACE2 via its 3′UTR.

## 2. Materials and Methods

### 2.1. Cell Culture Conditions

**Calu-3.** The lung-adenocarcinoma-derived Calu-3 epithelial cell line (ATCC^®^ HTB-55™, Burlington, ON, Canada) was cultured in minimum essential medium (α-MEM) supplemented with 10% fetal bovine serum, 1 mM L-glutamine, 100 units/mL penicillin and 100 µg/mL streptomycin, incubated at 37 °C in a humidified atmosphere containing 5% CO_2_ and passaged every 2 days.

**VeroE6:** Vero E6 cells (ATCC, CRL-1586™) were cultured in Dulbecco’s modified Eagle’s media (DMEM) supplemented with 10% fetal bovine serum, 1× L-glutamine, 100 units/mL penicillin and 100 µg/mL streptomycin and incubated at 37 °C in a humidified atmosphere containing 5% CO_2_.

**SARS-CoV-2 viruses.** The clinical isolates of SARS-CoV-2 (SARS-CoV-2/SB3) were purified from COVID-19-infected patients in Toronto, Canada and characterized previously [25]. The viral titers were determined using a TCID_50_ assay using Vero E6 cells. Experiments with SARS-CoV-2 were conducted with all precautions in a biosafety level 3 laboratory facility. All procedures were approved by the institutional biosafety committees at McMaster University. Calu-3 cells were infected with the SB3 isolate at an MOI of 1.0 in (biological) triplicate with SARS-CoV-2 as previously described [25]. Samples were collected at 24 h, 48 h and 72 h post transfection. Mock infections were included for each time point. The effectiveness of the infection was validated by qPCR and Western blot through the assessment of viral nucleocapsid mRNA and protein levels, respectively.

### 2.2. Protein Extractions and Western Blot

Cultured Calu-3 cells were lysed in RIPA buffer containing complete™ EDTA-free protease inhibitor cocktails (Sigma-Aldrich, Cat #R0278, Oakville, ON, Canada) and PhosSTOP™ phosphatase inhibitor (Thermo Scientific, Cat # 78446, Saint-Laurent, QC, Canada). Purified proteins were mixed with SDS loading buffer, denatured (10 min at 95 °C) and separated by SDS-PAGE (10% acrylamide gel); after which they were transferred to a PVDF membrane. Membranes were probed with the following primary antibodies: anti-tubulin (1/10,000; SCBT, Cat. #sc-5274, Dallas, TX, USA) and anti-SARS-CoV-2 nucleocapsid (1/2500; Invitrogen, Cat. # MA5-29981, Invitrogen, Burlington, ON, Canada) at 4 °C overnight, followed by incubation with the corresponding secondary antibodies. Chemiluminescent Western detection was performed with a C-digit instrument (LI-COR, Inc., Burlington, ON, Canada) and Clarity^TM^ Western ECL substrate reagents (Bio-Rad, Cat. #1705061, Hercules, CA, USA).

### 2.3. RNA Isolation

Total RNA was extracted from Calu-3 infected and non-infected cells at the 24 h, 48 h and 72 h time points using an RNA Easy kit (Qiagen, Cat. #74106, Toronto, ON, Canada) following the manufacturer’s recommendations. All RNA samples were treated with DNase I, quantified with the NanoDrop™ 2000 Spectrophotometer (Thermo Scientific™, Cat #ND-2000, Saint-Laurent, QC, Canada) and kept at −80 °C for subsequent experiments.

### 2.4. Illumina Nextseq Sequencing of Cells Infected with SARS-CoV-2 Viruses

Total RNA was shipped on dry ice to the ArrayStar sequencing platform (Rockville, MD, USA). Total RNA from each biological triplicate sample was used to prepare the miRNA sequencing library, which included the following steps: (1) 3′ adapter ligation; (2) 5′ adapter ligation; (3) cDNA synthesis; (4) PCR amplification and (5) size selection of ~130–150 bp PCR amplified fragments. The libraries were denatured as single-stranded DNA molecules, captured on Illumina flow cells, amplified in situ as clusters and finally sequenced for 50 cycles on Illumina Nextseq per the manufacturer’s instructions. The method, the experimental workflow and the flowchart of data analysis are detailed in Appendix A and our previous reports [26,27,28].

More than 6 million raw sequences were generated as clean reads from the Illumina Nextseq by real-time base calling and quality filtering. The adapter-trimmed reads above 16 nt varied between 41 and 69% on average (Appendix A). Based on Novoalign software, 27% to 54% of these adapter-trimmed reads (>16 nt) were aligned to known human pre-miRNA in miRBase21 (Appendix A) and their read length distribution in Calu-3 infected cells highlighted an abundance of 22 nt sequences generally corresponding to the average size of miRNAs (Appendix A). The status (infected vs. uninfected) and the stage (24 h vs. 72 h) of infection did not significantly alter the number of reads obtained (Appendix A).

For miRNA alignment, the maximum number of mismatches allowed was 1. When calculating miRNA expression, reads with counts less than 2 were discarded. miRNA expression levels were measured and normalized as transcripts per million (TPM) of total aligned miRNA reads miRNA read counts were used to estimate the expression level of each miRNA.

When comparing two groups of samples of profile differences (infected versus control), the “fold change” and *p*-value between each group are computed. miRNAs having fold changes ≥ 1.5, *p*-value ≤ 0.05 (unpaired) are selected as the DE miRNAs. The list of DE miRNAs is detailed in the Appendix A.

### 2.5. Go and KEGG Annotation/Enrichment

We used two databases to predict human (hsa) differentially expressed (DE) miRNAs’ target genes: Targetscan 7.1 (http://www.targetscan.org/vert_71, accessed on 1 September 2021) and mirdbV6 (http://mirdb.org/miRDB/, accessed on 1 September 2021) with the following parameters: species: hsa; score: ≥70 (miRdbV6); cumulative weighted context score: <−0.3; total context ++ score: <−0.3. The details of each prediction in the databases are in Appendix A. The filtered input was used to calculate over-represented biological pathways by following the recommended guidelines [29,30,31]. The miRNA–target interactions are experimentally validated by the database mirTarbase7.0. The ID of GO terms equals −log10 (*p*-value). Fisher’s exact test in Bioconductor’s topGO was used to find if there is more overlap between the DE list and the GO annotation list than would be expected by chance. The *p*-value produced by topGO denotes the significance of GO term enrichment in the DE genes.

### 2.6. RT-qPCR

**Detection of host mRNA targets and SARS-CoV-2 genome**. RNA extracted from host infected and non-infected cells were converted to cDNA with the HiFlex miScript II RT Kit (Qiagen, Cat. #218160, Frederick, MD, USA) following the manufacturer’s protocol. After dilution of the cDNA (1/10), qPCR was performed using the SSoAdvanced SYBR Green mix (Bio-Rad, Cat. #1725271, Hercule, CA, USA). The primers (Integrated DNA Technologies, Redwood, CA, USA; sequences listed in Appendix A), used at 1 µM final concentration, were designed with Primer-BLAST tools [32] and the best annealing temperature was chosen for each primer pair following a temperature gradient test. The qPCRs were performed using the StepOne™ Real-Time PCR System (Applied Biosystems, Cat. #4376357, Woburn, MA, USA) and data obtained (from StepOne™ Software, v2.3) were normalized to a reference gene (actin beta, ACTB) and reported to the controls (non-infected condition). The relative quantitation was calculated using the ddCt method [33].

**Detection of host miRNAs**. To monitor miR-1246 (GeneGlobe ID—YP00205630) and miR-1290 (GeneGlobe ID—YP02118634), we designed specific and sensitive (optimized with LNA technology [34]) microRNA primer sets for the miRCURY LNA miRNA Custom PCR Assay (Qiagen, Cat. #339317, Germantown, MD, USA). qPCR was performed using diluted cDNA (1/10) and the miRCURY LNA SYBR Green PCR Kit (Qiagen, Cat. #339346, Germantown, MD, USA) supplemented with ROX dye (Qiagen, Cat. #339346, Germantown, MD, USA). The qPCRs were performed using the StepOne™ Real-Time PCR System (Applied Biosystems, Cat. #4376357, Woburn, MA, USA). The U6 small nuclear RNA (RNU6) was used as a reference gene. Also, a UniSp6 RNA spike-in was used for cDNA synthesis and PCR amplification normalization as described previously [35].

### 2.7. Plasmid Constructs

The wild-type (WT) sequences of the ACE2 3′UTR (NCBI accession number: NM_001371415.1) and a mutated version were designed by using gBlocks^®^ gene fragments (Integrated DNA Technologies, Inc., Coralville, IA, USA). These sequences of 873 nt of the ACE2 3′UTR were introduced downstream of the Renilla luciferase (Rluc) reporter gene, the XhoI/NotI cloning sites of the psiCHECK2 vector (Promega, Madison, WI, USA). The details of the construction strategy have been previously described [35] and are summarized in Appendix A. All the constructs were independently confirmed by DNA sequencing at the Plateforme de Séquençage et Génotypage des Génomes (Centre de Recherche du CHU de Québec—CHUL, Québec, QC, Canada).

### 2.8. Cell Transfection and Dual-Luciferase Assay

Cell transfection and the dual-luciferase assay were performed as described previously with slight modifications [35]. Calu-3 cells were cultured in 6-well plates (300,000 cells per well) and transfected the following days at 70–80% with miRIDIAN miRNA-1246 and/or miRIDIAN miR-1290 mimics (Dharmacon, Horizon Discovery, Cat #C-301373-00-0002; Cat #C-301344-00-0002, Lafayette, CO, USA) and psiCHECK2 plasmids (WT or mutant). A total of 48 h after transfection, cells were washed with PBS and lysed with 500 μL of the passive lysis buffer (Promega, Cat. #E1980, Fitchburg, WI, USA). Luciferase activities were measured using the Dual-Luciferase^®^ Reporter Assay System (Promega, Cat. #E1980, Fitchburg, WI, USA) in a luminometer (TECAN INFINITE M1000 PRO), according to manufacturer’s instructions. Renilla luciferase (Rluc) expression was reported relative to the expression of the internal control firefly luciferase (Fluc). Rluc expression was further normalized to the control in which cells were co-transfected with an unrelated negative control miRNA mimic, referred to elsewhere as mock control. All assays were conducted in biological and technical triplicates in a 96-well format.

### 2.9. Statistical Analysis of qPCR Data

The statistical method used in each case is mentioned under figures. All statistical analyses were performed using GraphPad Prism version 9.3.1 (GraphPad Software, Inc., La Jolla, CA, USA), with statistical significance set at *p* < 0.05.

## 3. Results

### 3.1. SARS-CoV-2 Replication in Calu-3 Cells and Preliminary Analysis of the Molecular Mechanisms Underlying the Disease

To profile the miRNome of lung epithelial cells following SARS-CoV-2 infection, we infected Calu-3 cells with the clinical isolate SB3 [36]. Virus replication was confirmed by monitoring the SARS-CoV-2 nucleocapsid (N) transcript and protein expression levels. The SARS-CoV-2 nucleocapsid is an abundant viral RNA-binding protein essential for SARS-CoV-2 genome packaging and is highly expressed in infected cells [37]. The SARS-CoV-2 nucleocapsid mRNA and protein were specifically detected within infected cells by RT-qPCR and Western blotting, respectively (Figure 1). Between 24 h and 48 h post-infection, N mRNA levels were approximately the same, whereas in the late phase of infection (72 h), they increased eight-fold (Figure 1A). Meanwhile, N protein levels remained relatively stable over time (Figure 1B).

We also measured by qPCR a series of genes as an illustrative control of the infection efficiency. We performed a transcriptional analysis by qPCR to assess how SARS-CoV-2 modulates the expression of host genes that are involved in virus entry. We were interested in the SARS-CoV-2-mediated transcriptional modulation of key peptidases necessary for cellular entry, including ACE-2, TMPRSS2, ADAM-17 and Furin [6,38,39]. These proteases were all significantly upregulated at the early stage of infection (24 h, Appendix A) compared to mock-infected cells. Their expression level reached normal levels at the 48 h and 72h time points, except for ACE-2 and ADAM17, which were upregulated even in the late phase of infection (72 h).

We next monitored the expression of Interferon 1 beta (IFN1β) to assess the innate immune response and the levels of Interlukin-6 (IL-6) and C-X-C motif chemokine ligand 10 (CXCL10) to assess the levels of inflammatory cytokines. The expression of these genes is altered following SARS-CoV-2 infection [40,41]. As previously observed with peptidases, IL-6, IFN1β and CXCL10 were significantly upregulated at 24 h with a fold increase of about 2, then returned to normal levels at 48 h and 72 h, except for IFN1β which was elevated at 72 h (Appendix A). We also observed that the genes involved in fundamental cellular processes such as *SOCS4, AKT, NOTCH* and *WNT* [42,43,44,45]) were also upregulated in the early stages of infection (Appendix A).

In addition to being the main entry point identified for the virus into the cell, angiotensin-converting enzyme 2 (ACE2) is also a main player of the renin–angiotensin–aldosterone-system (RAAS) with angiotensinogen (AGT), angiotensin-converting enzyme (ACE) and renin [46].

Quantitative PCR analysis of these genes (Appendix A) showed a significant increase in ACE and AGT at 24 h, whereas renin was drastically downregulated (more than 50% reduction) at the same time. As with the previous data, neither of the aforementioned genes was significantly modulated at 48 h. At 72 h, renin, but not ACE or AGT, not only recovered its levels but was also upregulated 2-fold compared with baseline expression.

### 3.2. SARS-CoV-2 Infection Does Not Alter the Relative Abundance of miRNAs

We investigated how the miRNome, very often overshadowed by the general focus on the proteome, is modulated in the context of SARS-CoV-2 infection and how this modulation could explain our previous observations or might contribute to the widely described pathogenesis of SARS-CoV-2. Expression profiling of miRNAs at the 24 h and 72 h time points was performed in SARS-CoV-2-infected Calu-3 cells using RNA-Seq. Expression profiling of miRNAs (Figure 2) revealed that infection at the early or late stage has little or no effect in terms of their order of abundance. Specifically, the 20 most abundant miRNAs and their relative proportions were almost the same with or without SARS-CoV-2 infection (Figure 2).

### 3.3. A Finite Number of miRNAs Are Differentially Expressed upon SARS-CoV-2 Infection

As there was no significant change in the relative abundance of miRNAs, we were interested in the differentially expressed (DE) miRNAs, i.e., how many miRNAs were upregulated or downregulated and how many remained unaltered. When comparing the infected with non-infected samples (*n* = 3), we selected as the DE miRNAs those having fold changes greater than or equal to 1.5 and *p*-values less than or equal to 0.05.

Differences in miRNA expression upon SARS-CoV-2 infection were illustrated by a scatter plot (Figure 3), which displayed a very strong correlation (r = 0.9904 at 24 h and 0.9918 at 72 h) between miRNA profiles from both experimental conditions (infected vs. uninfected). At 24 h, 119 miRNAs were upregulated whereas 16 were downregulated. On the other hand, at 72 h, 87 miRNAs were upregulated whereas 26 were downregulated. The non-DE miRNAs totaled 1013 at 24 h versus 1154 at 72 h. When expressed as relative percentages (Appendix A), 88% and 91% were non-differentially expressed at 24 h and 72 h post-infection, respectively.

### 3.4. SARS-CoV-2 Infection Induces Consistent Upregulation of miR-1246 and miR-1290

To further consolidate our observations, the DE miRNAs were subjected to two additional analytical plots: volcano plots (Figure 4) showing statistical significance versus magnitude of change and heat maps (Figure 5) showing the hierarchical clustering of DE miRNAs. As shown by the correlation plot (Figure 3), the volcano plot (Figure 4) confirmed the very low number of miRNAs that were differentially expressed following SARS-CoV-2 infection (only 69 miRNAs versus 1079 non-DE at 24 h and 25 miRNAs versus 1242 non-DE at 72 h).

The heat map listing differentially expressed miRNAs from three independent samples (three infected, three uninfected controls) at 24 h and 72 h particularly highlights two miRNAs that are upregulated upon SARS-CoV-2 infection: miR-1246 and miR-1290 (Figure 5). Table 1 and Table 2 summarize the fold change along with the *p*-value and false discovery rate (FDR) of the top 10 up- and down-regulated miRNAs after infection at 24 h and 72 h, respectively. miR-1246, the most upregulated of Calu-3 miRNAs, showed a fold change of 14.55 (*p*-value = 0.0019; FDR = 0.1245) at 24 h and 6.23 (*p*-value = 0.0008; FDR = 0.4654) at 72 h post-infection (Table 1). miR-1290, which was ranked second, was upregulated 10.15-fold (*p*-value = 0.0016; FDR = 0.1245) and 5.39 (*p*-value = 0.0005; FDR = 0.4654, Table 1) at 24 h and 72 h, respectively. miR-196b-5p and miR-196a-5p, and miR-3924 and miR-30e-5p were the most downregulated at 24 h and 72 h, respectively (Table 2).

To confirm our observations and to validate our RNA-Seq data, we measured the expression level of miR-1246 and miR-1290 by qPCR on the same biological samples (Figure 6). Both miR-1246 and miR-1290 were found to be significantly upregulated at the two time points (24 h and 72 h). Following the same trend as in the RNA-seq data, miR-1246 showed higher fold change than miR-1290 (Figure 6). At 48 h, a condition not included in our RNA-Seq analysis, only miR-1246 appears significantly upregulated (1.56-fold).

### 3.5. miR-1246 Targets and Regulates ACE2

Ubiquitously expressed at varying levels, ACE2 is known to be the major cell entry receptor for SARS-CoV-2 [47,48]. The significant upregulation of miR-1246 led us to investigate whether it could affect ACE2 expression.

To assess this possible correlation, we used the dual-luciferase reporter assay and co-transfected Calu-3 cells with miR-1246 and a psiCHECK2 plasmid that carries a portion (approximately 900 base pairs, see details in Materials and Methods section) of the 3′UTR of ACE2 (wild-type or mutant) bearing inherent in silico predicted sites for miR-1246 (Figure 7). In the absence of the miR-1246 mimic, the normalized luciferase expression was reduced by about 15% both in the plasmid carrying the 3′UTR wild-type of ACE2 and the mutated version, suggesting the presence of elements with downregulatory properties in the 3′UTR of ACE2. When miR-1246 was added at increasing doses (25 to 100 nM), a 30 to 40% decrease in luciferase expression was observed for the plasmid with the ACE2 wild-type 3′UTR (Figure 7). On the other hand, the plasmid with the ACE2 mutated 3′UTR showed only a 10% decrease in luciferase signal. miR-1246 thus appears to be a direct and specific regulator of ACE2 mRNA. Similarly, we measured the expression of luciferase after the addition of miR-1290 alone (Appendix A) or in combination with miR-1246 (Appendix A). miR-1290 is the exact copy of miR-1246 without two adenine nucleotides on the 5′ side. However, it did not seem to induce any repression of luciferase expression (Appendix A). The mix of miR-1246 and miR-1290 did not trigger stronger repression of luciferase expression compared with miR-1246 alone (Appendix A). The contribution of miR-1290 was negligible, reinforcing the hypothesis of a specific interaction between miR-1246 and the 3′UTR of ACE2.

### 3.6. GO and KEGG Analyses of Differentially Expressed miRNAs

In an attempt to understand how the dysregulation observed at the level of miRNAs in host cells would contribute to SARS-CoV-2 pathogenesis, symptoms or host defense against SARS-CoV-2 infection, we subjected the above DE miRNAs to KEGG and GO analysis.

“RAS and MAPK signaling pathway”, the “cholinergic synapse” and “endocytosis”, were among the significantly enriched KEGG pathways related to genes targeted by the upregulated miRNAs (Figure 8A), in both early (24 h) and late stages (72 h) of infection. Some KEGG pathways were found uniquely at 24 h (“aldosterone synthesis and secretion”) or 72 h (“neurotrophin signaling pathway”).

“RAS and MAPK pathways” were also found in the significantly enriched KEGG pathways of downregulated miRNAs but with half as many genes involved (Figure 8B). “PI3K-AKT, AMPK, AGE-RAGE signaling pathways” and “cell cycle” were among the significantly and specifically enriched KEGG pathways related to genes targeted by downregulated miRNAs at 24 h. “Renin secretion”, “regulation of actin cytoskeleton” and “glutamatergic synapse” were among the enriched KEGG pathways at 72 h.

Furthermore, using the GO knowledge base (biological process, cellular component and molecular function) we identified the main pathways in which DE miRNAs may be involved (Appendix A). Upon SARS-CoV-2 infection, the upregulated miRNAs seemed to affect several biological processes, we mention, among others, “regulation of cellular processes and localization” and “signaling”; the cellular components such as “intracellular” and “cytoplasm”; and the molecular function such as “protein binding” and “transcription factor activity” (Appendix A). For the downregulated miRNAs, among other terms, the three GO aspects emphasized “metabolic process” for the biological process, “intracellular” for the cellular components and “protein binding” for the molecular functions (Appendix A).

## 4. Discussion

In this study we investigated COVID-19 from the perspective of noncoding RNAs, in particular miRNAs. The challenge was, as in our previous works on the HIV/AIDS virus [49,50], the Ebola virus [26,35] and the Hepatitis B virus [51], to understand how SARS-CoV-2 modulates the host miRNA profile during infection and eventually whether it encodes expressed viral miRNAs involved in pathogenesis.

We used Calu-3, a common, sensitive and effective preclinical model for studying human respiratory processes and lung-injury-related diseases [52]. Calu-3 is a well-established in vitro model of respiratory virus infection, an alternative approach that mimics the characteristics of bronchial epithelial cells very well [53,54]. However, Calu-3 is a lineage derived from lung adenocarcinoma. It would be interesting to perform a further test of the experimental strategy in normal lung epithelial cells.

Our RNA-Seq data and the subsequent bioinformatics analyses enabled us to identify more than 1000 host miRNAs but none from the viral genome (using a pipeline similar to the one we used for Ebola [35]). Compared with the uninfected controls, the volcano plot showed that 64 and 22 host miRNAs were upregulated at 24 h and 72 h, respectively, whereas only 5 and 3 host miRNAs were downregulated at 24 h and 72 h, respectively, in the infected Calu-3 cells. Our results suggest that SARS-CoV-2 induces an early and transient response of host genes involved in global regulation consistent with our previous observations [25].

Similar to our studies on Ebola virus infection [26], very few miRNAs were modulated by the presence of the virus, the vast majority remaining non-differentially expressed. This phenomenon could also be the result of an evolutionary adaptation of viruses allowing them to evade miRNAs [15].

In a previous study conducted by a team in Toronto in the human lung epithelium 24 h after SARS-CoV-2 infection, the analysis of differentially expressed miRNAs revealed that only 45 miRNAs were differentially expressed [23]. In the peripheral blood from ten COVID-19 patients, Li et al. found a total of 73 human miRNAs to be differentially expressed [21]. Other studies involving other viruses also reported similar findings [55,56].

In peripheral blood samples, it was observed that miR-16-2-3p (upregulated) and miR-627-5p (downregulated) were the ones most highly regulated [21]. A similar (cross-sectional) study instead highlighted other miRNAs such as miR-17-3p as the most up-regulated in correlation with disease progression and miRNAs such as hsa-miR-31-3p, hsa-miR-29a-3p and hsa-miR-126-3p as the most repressed. In our study, these ranks are respectively occupied by miR-1246 (upregulated) and miR-196b-5p (downregulated).

The variation in the list of miRNAs obtained could be explained by the difference in approaches. The nature of the samples used (human patient samples vs. human cell cultures), the infection parameters such as MOI (low vs. high) and time course (hours vs. days) as well as the experimental RNA-Seq analysis strategy (from extraction of RNAs to the bioinformatics analysis pipeline) can all affect the pattern of microRNAs in one way or another. Nevertheless, these data, in their context, all provide a better understanding of the virus’s intimate molecular biology at another level of complexity.

Consequently, in the context of SARS-CoV-2 infection or RNA viruses such as Ebolavirus, there seems to be a general tendency for specific, targeted and pool-limited regulation of miRNAs. SARS-CoV-2 may use pathways that minimally affect miRNA-related cellular processes. From another perspective, the susceptibility of the lung epithelia to infection could arise from the lack of an important miRNA-associated protective mechanism [23,57]. miRNAs appear to be more involved in pathways conferring robustness to biological processes than in defense against foreign nucleic acid [58]. Further studies are warranted to clarify the role of DE miRNAs; however, they may be pivotal in the control of viral tropism [59] and may serve as potential informative biomarkers [60]. Furthermore, we observed that miR-196(a/b)-5p, one of the top 10 most significantly repressed microRNAs during the early stage (in our study, 24 h post-infection), has been reported in previous studies to be potentially capable of interacting directly with the viral genome, specifically via ORF1(a,b) and ORF2 (corresponding to the Spike protein) [61]. The repression of such an miRNA might be part of the viral strategies for better replication and escape from the host’s defenses.

GO and KEGG analyses show that essential pathways are affected by the DE miRNAs upon SARS-CoV-2 infection. This was the case for RAS and MAPK signaling pathways, which are involved in all the fundamental processes of the cell, such as transcription, differentiation, proliferation, migration and survival [62]. RAS is regulated by the let-7 miRNA family [63], which is among the 20 most expressed miRNAs in Calu-3 and among which five miRNAs were significantly downregulated at 24 h. MAPK is also controlled by miR-4728 [64], which was among the three most upregulated miRNAs at 24 h and 72 h. Additionally, the most upregulated miRNAs, such as miR-1246 and miR-1290, have been reported to be potentially involved in fundamental cellular processes [65,66].

Among other pathways targeted by the upregulated miRNAs, we found cell endocytosis [67] and cholinergic pathways [68], which are of fundamental relevance to the infection process. In the same vein, other downregulated miRNAs are also associated with genes involved in the core of cell survival mechanisms. These include AMPK and PI3K/AKT signaling pathways, which are key energy regulators in the cell and participate in host immune function [69,70]. Strategies of the virus versus the host are not so easily distinguished in the modulation of miRNAs; indeed, the overall regulation is likely a combination of co-evolving virus–host strategies. Besides sharing common targets and interacting with each other [71], each miRNA can have an extended influence on the regulation of mRNAs and their evolution (especially on their 3′UTRs) [12]. This may explain the lack of specificity in GO and KEGG analyses of miRNA target genes.

In this study, we explored and confirmed the interactions between upregulated miR-1246 and ACE2 mRNA in Calu-3 cells. A previous study on pulmonary microvascular endothelial cells (PMVECs) reported miR-1246 represses ACE2 expression by binding to the 3′UTR [72]. By targeting ACE2, miR-1246 mediates pulmonary endothelial cell apoptosis, acute lung injury and acute respiratory distress syndrome (ARDS) [72]. Furthermore, patients with chronic obstructive pulmonary disease or pulmonary hypertension [73,74,75] demonstrate low miR-1246 expression and high ACE2 expression compared with healthy subjects. Inversely, in relation to ACE2, increased levels of miR-1246 were reported to associate with acute respiratory distress syndrome (ARDS) [18,72,76]. miR-1246 therefore appears to be a specific and complex biomarker for lung diseases and could be critically informative to the risk of COVID-19 complications.

Given the bioinformatics predictions [76], the experimental validation on two cell types (Calu-3: this study, PMVECs [72]) and the specificity of the interaction, miR-1246 may emerge as a prime “therapeutic” target to modulate ACE2. Although the role of ACE2 is ambiguous in the progression of COVID-19 [77], there is no doubt that its availability remains the major tropism determinant for SARS-CoV-2 [74], and miR-1246 is a game changer in ACE2 provision. In the context of SARS-CoV-2 infection, our study provides the evidence of a direct interaction between ACE2 and miR-1246 (14 times upregulated vs. control). Considering the involvement of miR-1246 in numerous regulatory pathways (regulates activity of RAF/MEK/ERK, GSK3β, Wnt/β-catenin, JAK/STAT, PI3K/AKT, THBS2/MMP and NOTCH2 pathways, [65]), it would be interesting to explore this association to grasp the complexity of their relationship and the consequences of the interaction in disease progression.

Our qPCR results indicated an upregulation of ACE2 at 24 h and 72 h but not at 48 h. Same samples, analyzed by RNA-Seq, showed an upregulation of miR-1246 at 24 h and 72 h. This suggests that the repressive effect of miRNA on ACE2 does not occur spontaneously and that it does not necessarily lead to the degradation of the ACE2 mRNA. Indeed, there is current evidence highlighting multiple, non-canonical modes of miRNA-mediated mRNA regulation [78]. In experiments involving ACE2, it is important to consider the dual impact of this protein—sometimes an ally for immune defense, sometimes a viral entry point for SARS-CoV-2 [79]. Is the increased ACE2 the result of the effect of the viral attack or of the immune defense? The mechanisms underlying this balance are not well understood. In parallel, the increase in miR-1246 expected to counteract ACE2 is not paradoxical since these measurements are at the post-transcriptional level at a given time and do not predict the post-translational scenario for ACE2.

It is also noteworthy that the dual-luciferase experiment is an “embedded system” where we do not have all the properties of the 3′UTR of ACE2 since the transcript is more than 1500 nt whereas only 750 nt are present on the psiCHECK plasmid in the experiment. The entire 3′UTR or ideally the entire transcript would have better simulated what happens in vivo with all folding scenarios (secondary structures exposing or hiding potential binding sites). It is not excluded that ACE2 may have several other microRNA response elements/MREs located beyond the 3’UTR, either in the 5′UTR or in the coding region.

We assessed the expression levels of numerous transcripts involved in the pathogenesis of SARS-CoV-2. At the transcriptional level, we attempted to explore the correlation between RAS and SARS-CoV-2, which is being extensively studied [80] but is still poorly understood. The RAS dysregulation observed during SARS-CoV-2 infection is believed to contribute to adverse cardiovascular and respiratory effects, hypercoagulation and inflammation [81]. First, we found an early and drastic drop in the level of renin, an upstream player in the RAS enzymatic cascade, suggesting a possible slowdown effect against the viral infection [82]. However, the observed increase in the ACE/ACE2 ratio at 24 h seems to be a negative viral effect and is known to be a risk factor for worse outcomes in COVID-19 infection [83]. At 72 h, the joint augmentation of renin and ACE2 could portend a favorable scenario since both actors direct the protective pathway of the RAS system [81]. The accumulation of angiotensinogen, the substrate for the formation of angiotensin I, seemed to be a direct result of the renin decrease.

Through its peptidases and interacting receptors, RAS also represents an integrated inflammatory system [84]. We observed an early and transient upregulation of pro-inflammatory biomarkers of signaling pathways that participate in the cellular response triggered by viral infection [85] including CXCL10, whose increased levels are usually associated with an urgent transfer to the ICU (intensive care unit) or with death [86], and the prime candidate for mediating inflammation in COVID-19, IL-6 [84]. At 72 h, the increase in IFN1β, with its antiviral and immunomodulatory effects [85], seems to overcome the pro-inflammatory actors upregulated at 24 h (CXCL10, IL-6). It should be noted, however, that interferons do not improve outcomes (adverse symptoms and consequences) for hospitalized adults with COVID-19 [86].

However, the above qPCR results only reflect what happens at the transcript level and do not provide information on the expression level of the proteins under these experimental conditions. Further analyses are needed to elucidate the enigmatic role of the enzymes ACE2 and ACE as well as renin in the RAS system in the context of COVID-19. It would also be interesting to see the differential contribution of different MOIs (low and high) on the miRNA expression profile.

In conclusion, our study documents new aspects of the pathogenesis of SARS-CoV-2 with the purpose of further reinforcing the available clinical and molecular data. Contradictory results about the biological significance of biomarkers, treatments and other clinical parameters related to COVID-19 were observed throughout the pandemic [84].

## Figures and Tables

**Figure 1 viruses-15-00496-f001:**
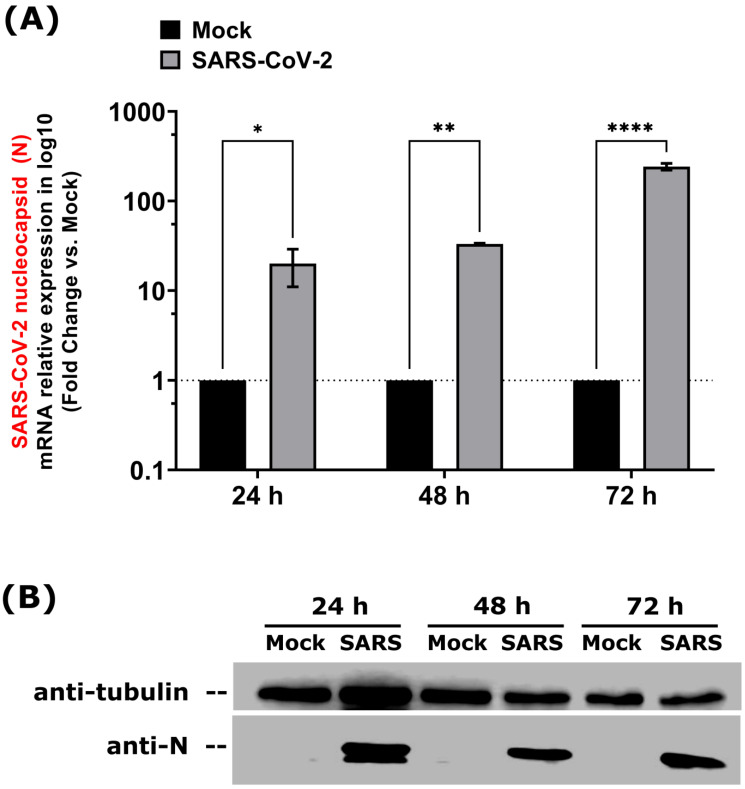
SARS-CoV-2 replication in Calu-3 cells. SARS-CoV-2 *nucleocapsid* gene (N) expression was monitored upon SARS-CoV-2 infection in Calu-3 cells by RT-qPCR (**A**) using specific primers and Western blot (**B**) using anti-N (42kDa) and anti-Tubulin (55 kDa, control) antibodies. qPCR data were normalized to a reference gene (Actin beta, *ACTB*), reported to mock (unrelated negative control) and expressed using a relative quantitation method (ddCT). **Statistical analysis.** All data (**A**) presented were calculated from three biological replicate (*n* = 3) measurements ± SD. The ordinary two-way analysis of variance (ANOVA) and Šídák’s multiple comparisons test were used for statistical analysis. Statistically significant differences (fold change vs. mock) are indicated by stars (*): * *p* < 0.05; ** *p* < 0.01; **** *p* < 0.0001.

**Figure 2 viruses-15-00496-f002:**
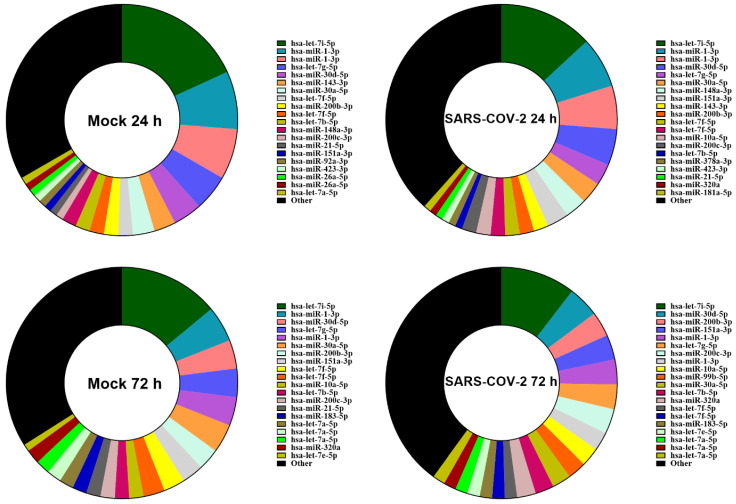
miRNA expression profiling of Calu-3 cells infected or not with SARS-CoV-2 virus. Circular diagram representing the 20 most abundant microRNAs in each experimental condition (Mock- or SARS-CoV-2-infected; 24 h and 72 h) along with other microRNAs and their relative proportions assessed by RNA-Seq. The ranking was based on the normalized tag number of miRNAs averaged from 3 biological replicates.

**Figure 3 viruses-15-00496-f003:**
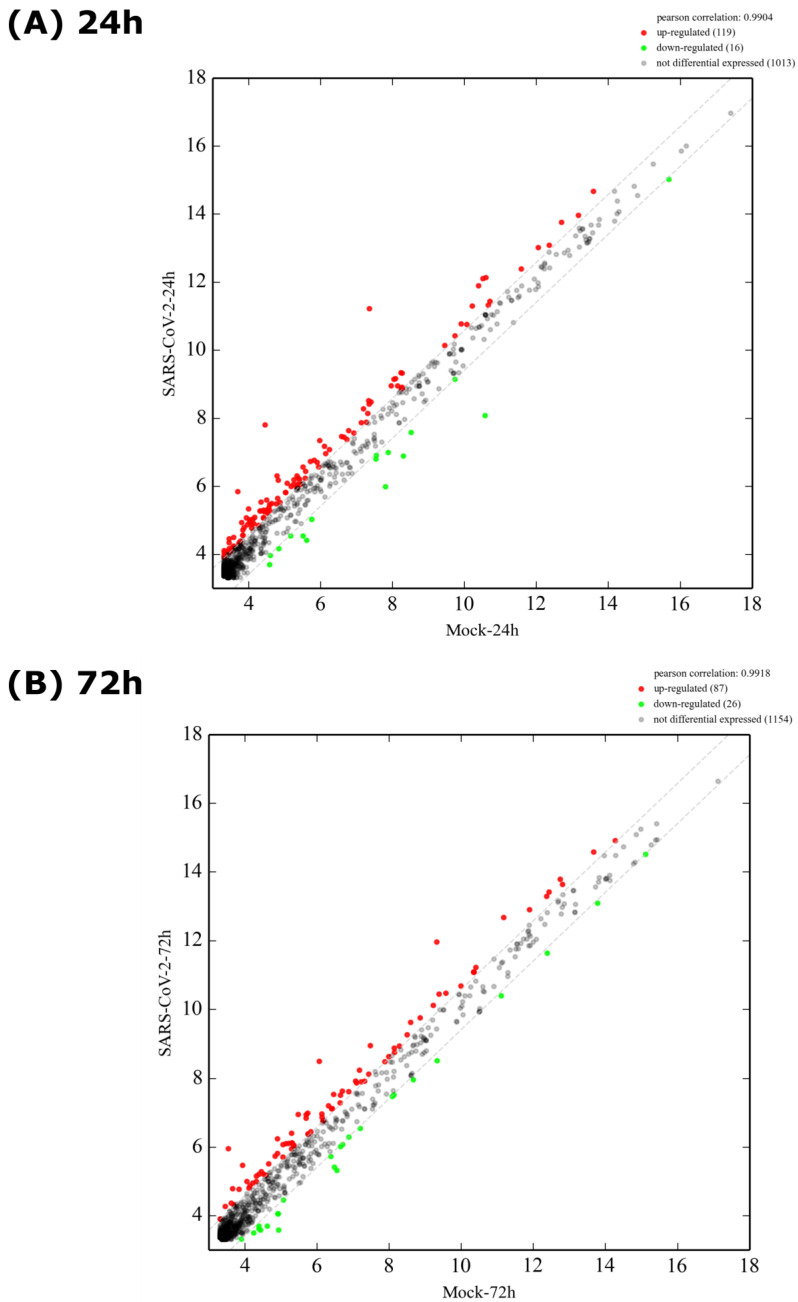
Scatter plot of differentially expressed miRNAs of Calu-3 cells infected or not with SARS-CoV-2 virus. Scatter plot showing up-regulated (red spots), downregulated (green spots) and non-differentially expressed miRNAs (black spots) at (**A**) 24 h and (**B**) 72 h post-infection.

**Figure 4 viruses-15-00496-f004:**
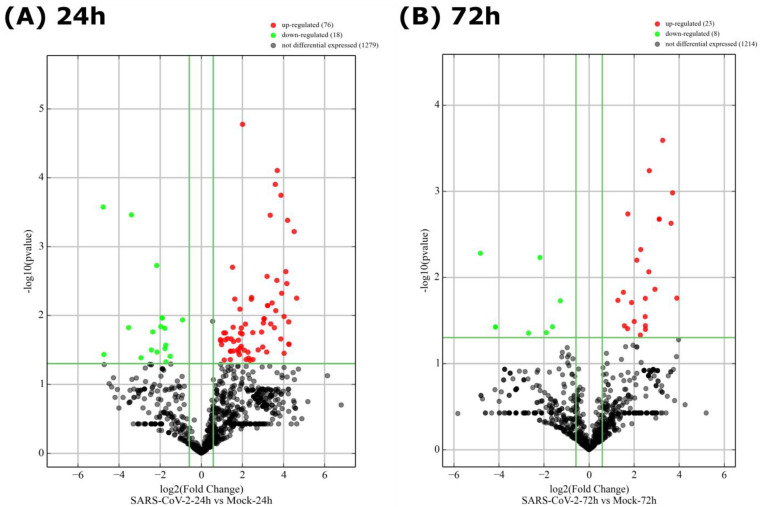
Volcano plot of differentially expressed miRNAs. Plots depicting the magnitude (log 2 reporting fold change, FC, x-axis) and significance (−log 10 adjusted *p* value, y-axis) of differentially expressed miRNAs between Mock and SARS-CoV-2-infected Calu-3 cells (**A**) 24 h and (**B**) 72 h post-infection. The green, red and black spots represent upregulated, downregulated and non-differentially expressed miRNAs, respectively. The statistical significance threshold (*p* ≤ 0.05) is illustrated by the horizontal green line and the threshold of fold change (log 2 FC > 1 or <−1) by the two vertical green lines.

**Figure 5 viruses-15-00496-f005:**
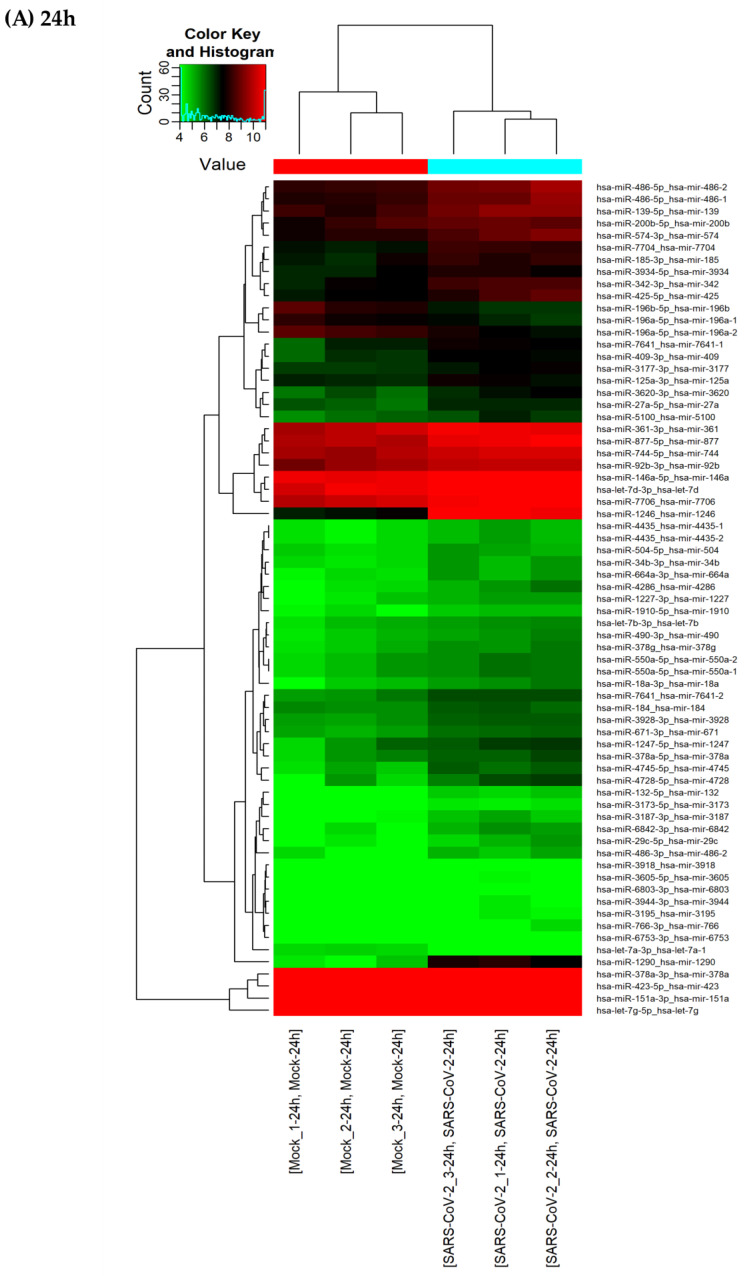
Heat map and hierarchical clustering of differentially expressed miRNAs. miRNA expression levels are illustrated using a histogram and color key. The upper dendrogram illustrates the clustering of samples (*n* = 3) infected or not (Mock) with SARS-CoV-2 at (**A**) 24 h and (**B**) 72 h post-infection. The left dendrogram illustrates the clustering of miRNAs differentially expressed and listed in the left vertical axis. Each row represents a miRNA, and each column represents a sample.

**Figure 6 viruses-15-00496-f006:**
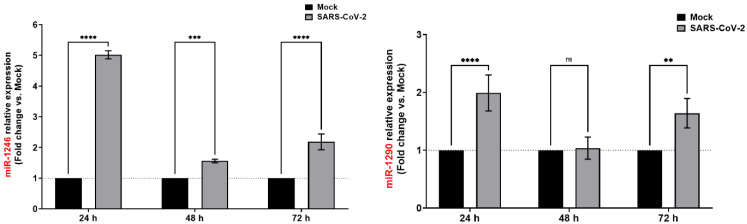
Validation of miR-1246 and miR-1290 upregulation (RNA-Seq data) by qPCR. miR-1246 and miR-1290 were monitored by RT-qPCR upon SARS-CoV-2 infection in Calu-3 cells. qPCR data were normalized to a reference gene (U6 snRNA), reported to mock and expressed with a relative quantitation method (ddCT). **Statistical analysis.** All data presented were calculated from three biological replicate (*n* = 3) measurements ± SD. The ordinary two-way analysis of variance (ANOVA) and Šídák’s multiple comparisons test were used for statistical analysis. Statistically significant differences (fold change vs control) are indicated by stars (*): ** *p* < 0.01; *** *p* < 0.001; **** *p* < 0.0001. ns, nonsignificant.

**Figure 7 viruses-15-00496-f007:**
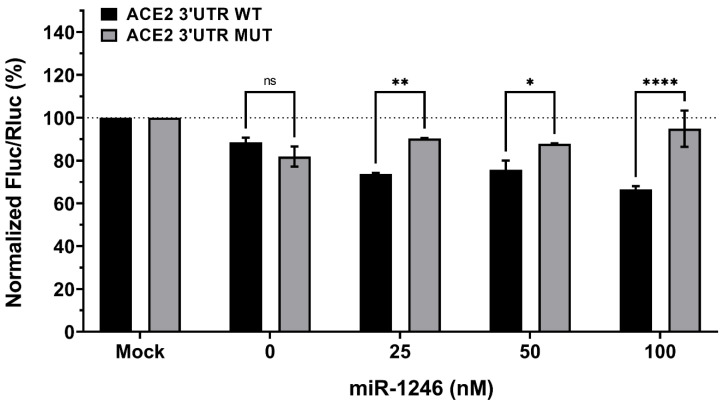
Modulation of human ACE2 mRNA by hsa-miR-1246. Calu3 cells were co-transfected with Homo sapiens (hsa) human miR-1246 mimic (0, 25, 50 and 100 nM) and a psiCHECK2 reporter construct (50 ng; see Appendix A), in which the Rluc reporter gene was coupled with the wild-type (WT) or mutated (MUT) human ACE2 3′ untranslated region (UTR). An unrelated, negative miRNA control (Mock) was used for normalization in addition to the internal normalizer Fluc. The concentration “0 nM” corresponds to the transfection-reagent-only control. Statistical analysis: Data were calculated from three biological replicates and expressed as means ± SD. The two-way analysis of variance (ANOVA) and Šídák’s multiple comparisons test were used. Statistically significant differences (fold change WT vs. MUT) are indicated as follows: * *p* < 0.05; ** *p* < 0.01; **** *p* < 0.0001; ns, nonsignificant.

**Figure 8 viruses-15-00496-f008:**
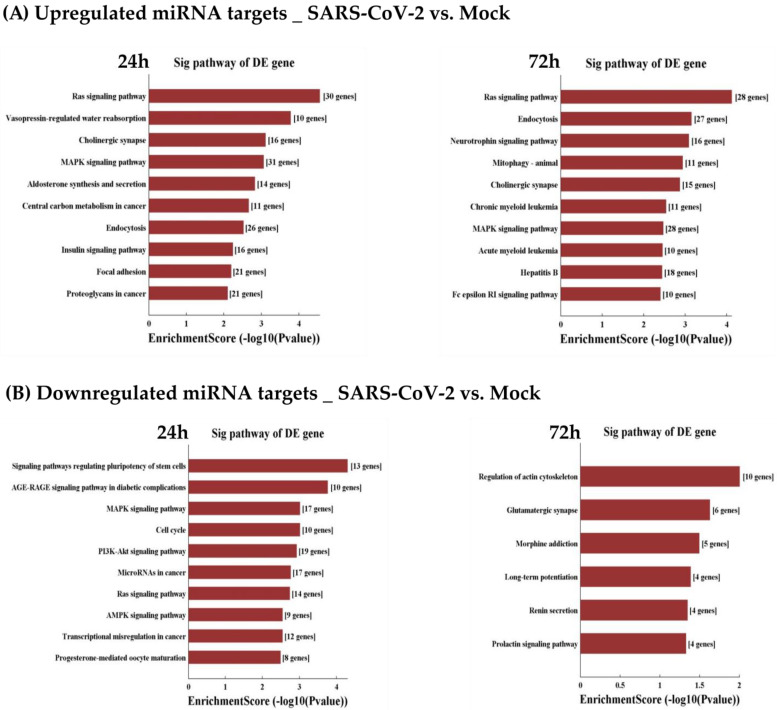
Enrichment score values of the top 10 enrichment pathways following SARS-CoV-2 infection vs. Mock. The bar plot shows the top 10 significantly enriched KEGG pathways of genes targeted by (**A**) upregulated and (**B**) downregulated miRNAs at 24 h and 72 h post-infection.

**Table 1 viruses-15-00496-t001:** Top 10 upregulated miRNAs in Calu-3 cells at 24 h and 72 h post-SARS-CoV-2 infection, compared with uninfected control (Mock).

Mature miRNAs	Fold Change	*p*-Value (Unpaired)	FDR
SARS-CoV-2-24 h vs. Mock-24 h			
hsa-miR-1246	14.550813	0.001936356	0.124553971
hsa-miR-1290	10.151515	0.001616931	0.124553971
hsa-miR-4728-5p	2.855422	0.032727219	0.340049232
hsa-miR-4745-5p	2.552941	0.002060005	0.124553971
hsa-miR-6842-3p	2.520833	0.00909142	0.246889589
hsa-miR-4286	2.278689	0.040817409	0.366081138
hsa-miR-425-5p	2.263918	0.036130815	0.348556098
hsa-miR-29c-5p	2.195652	0.047218269	0.408413103
hsa-miR-3187-3p	2.190476	0.018900103	0.293985775
hsa-miR-486-5p	2.147124	0.031260241	0.340049232
SARS-CoV-2-72 h vs. Mock-72 h			
hsa-miR-1246	6.234131	0.000859555	0.465454769
hsa-miR-1290	5.393035	0.000599562	0.465454769
hsa-miR-4728-5p	2.768657	0.032469057	0.605339181
hsa-miR-5100	2.522222	0.038432487	0.605339181
hsa-miR-7977	2.098113	0.043036275	0.605339181
hsa-miR-574-3p	2.036113	0.015325686	0.605339181
hsa-miR-194-3p	1.897764	0.048234042	0.605339181
hsa-miR-877-5p	1.861196	0.007521499	0.605339181
hsa-miR-1306-5p	1.860465	0.046973589	0.605339181

FDR = False Discovery Rate.

**Table 2 viruses-15-00496-t002:** Top 10 downregulated miRNAs in Calu-3 cells at 24 h and 72 h post-SARS-CoV-2 infection, compared with uninfected control (Mock). No other miRNAs meeting the criteria other than the ones listed in this table were identified. Therefore, the top 10 does not contain 10 miRNAs.

Mature miRNA	Fold Change	*p*-Value (Unpaired)	FDR
SARS-CoV-2-24 h vs. Mock-24 h			
hsa-miR-196b-5p	0.374473	0.036770814	0.351774119
hsa-miR-196a-5p	0.521818	0.01962821	0.293985775
hsa-miR-196a-5p	0.540311	0.023390933	0.308652771
hsa-let-7g-5p	0.633044	0.008259531	0.246889589
hsa-let-7a-3p	0.643836	0.000312666	0.05127724
SARS-CoV-2-72 h vs. Mock-72 h			
hsa-miR-3924	0.587302	0.011169221	0.605339181
hsa-miR-30e-5p	0.609796	0.040388915	0.605339181
hsa-miR-145-3p	0.642384	0.003165844	0.605339181

FDR = False Discovery Rate.

## Data Availability

All raw small RNA-seq data generated in this study have been submitted to the NCBI Gene Expression Omnibus under accession number: BioProject accession: PRJNA901149; GEO accession: GSE217863 Access link: https://www.ncbi.nlm.nih.gov/geo/query/acc.cgi?acc=GSE217863.

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
