# Peer review of "Altered microRNA Transcriptome in Cultured Human Airway Cells upon Infection with SARS-CoV-2"

_viruses, 2023, doi:10.3390/v15020496_

Round 1
Reviewer 1 Report (New Reviewer)
The authors evaluated the microRNA levels of cultured lung epithelial cells infected by SARS-CoV-2 which seems to be valuable. This paper also discovered top pathways related to the deregulated miRNAs to determine the relationship between these deregulated miRNAs and their target mRNAs during COVID-19 infection. The manuscript is well written. The comments from this reviewer on this work are as follows.
1. It is stated that viral infection was induced in the Calu-3 cell line, a non-small-cell lung cancer cell line. Please explain why you select these cells and why did not use normal lung epithelial cells since the pattern of microRNAs can be altered in cells compared to controls.
2. In this study, any changes in miRNA profiling were evaluated at 24, 48 and 72 hours post-infection. The dose dependence study is lacking in this study; please make it clear to readers why you used a single load (an MOI of 1.0) of viruses. Please explain why you did not consider it or it can be suggested as a limitation of the study.
3. In figure 1, the size of the protein is missing. Please revise it.
4. Please kindly increase the resolution of Figure 5.
5. I noticed that you have mentioned a heat map in the figures. If any image (heat map) is created using software/database or any other software/database is used apart from the above-mentioned software, then kindly mention them and please provide the written permission and citation of that software/database (if required by the software/database website).
6. The authors cited supplementary Figure S1-10; however, I could not find them in the supplementary folders.
7. Please separate the Tables’ titles from abbreviations. They should come at the bottom of the related Tables.
8. The article would require being revised significantly to improve the grammar, typos, and ...
“In addition, we also”
Author Response
Please see the attachment.

Reviewer 2 Report (New Reviewer)
In this study, the authors performed a two-time point of miRNome analysis using a physiologically relevant cell line Calu-3 with or without SARS-CoV-2 infection. Interestingly, they found that after 24-hour infection, 119 miRNAs were upregulated while 16 were downregulated. After 72-hour infection, 87 miRNAs were upregulated while 26 were downregulated. These findings showed a larger number of miRNAs that are differentially expressed before and after SARS-CoV-2 infection than those two previous papers as the authors mentioned in the Discussion section. The miRNome are less studies compared to RNASeq or proteome in SARS-CoV-2 infection, which is the significance of this current study. The observation is intriguing. However, the quality of this study is poor and does not match the standard of viruses unless the major concerns listed below are well addressed.
Major concerns:
1. The novelty of this study is limited. Previous studies have been performed to reveal the difference of miRNomes before and after SARS-CoV-2 infection and try to find a novel prognosis marker and drug target for COVID-19 treatment. So this manuscript is not the first one to show the miRNome after SARS-CoV-2 infection. Since miRNome plays an important role in regulation protein expression, it is important to investigate the changes of miRNome before and after SARS-CoV-2 infection. The authors showed a novel and larger dataset of miRNAs than before, however, the differentially expressed miRNAs are less characterized in the current manuscript, which is a big concern. ACE2 mRNA level was tested under treatment of the miR-1246 mimic. More miRNAs that were upregulated on the list after infection should be tested. Also, not only for ACE2, the mRNA level of more proteins (including TMPRSS2, ADAM17, Furin, IL-6, IFN1b,etc) should be tested using miRNA mimics, as the authors predicted that these proteins may be affected by miRNAs.
2. To further support the major findings in the manuscript, it will be good to check the expression level of proteins of interest after treatment of miRNA mimics.
Minor concerns:
1. The authors should comment on more previous miRNome papers and point out the potential reasons why the miRNA list varies so much in the current paper and previous ones.
2. It will be interesting to see whether the miR-1246 mimic treatment affects the viral entry of authentic SARS-CoV-2 viruses.
Round 2
Reviewer 1 Report (New Reviewer)
Please accept the changes (with the track change) in the supplementary Tables.
Author Response
Please see the attachment.

Reviewer 2 Report (New Reviewer)
If the authors can try to test either the mRNA level of more proteins (including TMPRSS2, ADAM17, Furin, IL-6, IFN1b,etc) by miRNA mimics or the effect of miR-1246 mimic on the cell entry of authentic viruses, the manuscript will be much improved. It seems the authors do not want to do more experiments due to the funding limit. If so, the authors should list how many and which miRNAs are reported by previous studies and which ones are new from the current study. After the comparison, I am fine that the manuscript will be accepted for publication in VIURSES.
Author Response
Please see the attachment.

This manuscript is a resubmission of an earlier submission. The following is a list of the peer review reports and author responses from that submission.
Round 1
Reviewer 1 Report
Diallo and colleagues aims to identify changes in microRNA expression upon SARS-CoV-2 infection. For this study authors infected the CaLu-3 cell line and collected cells at 24 h, 48 h, and 72 h post transfection for miRNA sequencing analysis. Results and analysis of the sequencing results are presented.
Changes in microRNA expression may be of importance in disease pathogenesis, thus the results of this study are of great interest, however there are several issues that need to be addressed before the study is ready for publication.
Major comments:
1. Detailed methods describing statistical analysis are missing or inadequate.
For instance how was DE analysis performed? Similarly, it is unclear how KEGG and GO analyses for miRNA was performed. We are not aware of individual miRNA being included to a great extent in current gene sets/pathways. Did the authors use predicted targets to identify enriched gene sets and or pathways? If predicted targets were used, is there filtering of high-confidence targets (this would reduce noise in data). If taking all targets, many/all pathways are going to show up.
The statistical tests used in the DE analysis need to be appropriately justified. For instance, the authors use a parametric test (unpaired t- test) for read count data, however read counts are not normally distributed. Also it is inappropriate to use normalized read counts for DE analysis since you lose the dynamic range. Should consider using tools such as edgeR or DEseq or any statistical test that assumes negative binomial distribution.
2. The inclusion of the sections entitled "SARS-CoV-2 Infection May Stimulate Peptidases Involved in Virus Entry", "SARS-CoV-2 infection may lead to prompt activation of the innate immune response and triggers fundamental cellular processes", "SARS-CoV-2 infection modulates the Renin Angiotensin Aldosterone System (RAAS)" seem out of the scope of the study that aims to study the altered microRNA transcriptome. Without better justification, it is recommended that these sections be removed or placed in supplementary data as controls indicating effective infection as is already demonstrated in section 3.1.
Minor Comments:
- Western blot in 1B is over-saturated, thus hard to interpret. Double band at 24h?
- Regarding, "Their expression level reached normal levels at the 48 h and 72h time points, except for ACE-2 and ADAM17 which remained upregulated even in the late phase of infection (72 h)." The use of word "remained" is misleading here because 48h point wss not significant.
- Same issue here: "...IL-6, IFN1β and CXCL10 were significantly upregulated at 24 h with a fold increase of about 2, then returned to normal at 48 h and 72 h except for IFN1β which remained elevated at 72 h (Figure 3)."
- Can the authors measure ACE2 protein expression?
Reviewer 2 Report
Here, Diallo et al. explored the effects of SARS-CoV-2 infection on the phenotype of human airway epithelial cells. They found that the infection upregulates the expression of host carboxypeptidase involved in the shedding of ACE2 ectodomain and mRNAs involved in the innate immune response and RAAS. Then, they performed a small RNAseq to investigate miRNAs differentially expressed in infected cells and the ability of DE miRNAs to modulate the expression of ACE2 in infected cells.
While the topic is certainly worth investigation, and the study design is appropriate, I have some major concerns regarding data presentation and interpretation, which could compromise the main message of the study:
1. No methods were provided for the statistical analysis of RNAseq data. Which software was used for the analysis? Did the authors perform a prefiltering?
2. The RNAseq dataset has not been deposited in a public repository. This task takes a little time and guarantees the integrity of the data.
3. Table 1 shows that the authors considered the DE miRNAs on a p-value basis, instead of FDR. In RNAseq experiments, this is inappropriate, since, given the high number of variables, the probability of false positives is very high and should be accounted for. In this case, the authors could have considered an arbitrary FDR threshold, such as FDR<0.20, to include the two miRNAs they were interested in. Or, additionally, validate a greater number of miRNAs from their list. However, the pathway analysis would have been unreliable, since it is based on all miRNAs the authors deemed to be differentially expressed. The authors should justify their methodology, as this can represent a major flaw.
4. Interpretation of the interaction between miR-1246 and ACE2 is quite underwhelming. The authors found an upregulation of ACE2 and miR-1246 in infected cells. Using a luciferase experiment, they demonstrated that miR-1246 binds to the 3'UTR of ACE2. Thus, degradation of ACE2 mRNA would be expected. This is only briefly discussed in page 19, where the authors state that "current evidence highlighting multiple, non-canonical modes of miRNA-mediated mRNA regulation " exists. In my opinion, they should expand on this aspect and demonstrate that this non-canonical modulation actually occurs in their experiment. Did they test ACE2 protein expression in these cells? Did they explore a more plausible interaction between some of the downregulated miRNAs (which were not considered) and ACE2?